# The Association between Specimen Neuromuscular Characteristics and Urinary Incontinence after Robotic-Assisted Radical Prostatectomy

**DOI:** 10.3390/diagnostics14182001

**Published:** 2024-09-10

**Authors:** Tomer Bashi, Jonathan Margalioth, Ziv Savin, Ron Marom, Snir Dekalo, Ibrahim Fahoum, Rabab Naamneh, Roy Mano, Ofer Yossepowitch

**Affiliations:** 1Department of Urology, Tel Aviv Sourasky Medical Center, Tel Aviv 6423906, Israel; jmyakada@gmail.com (J.M.); zivsavin23@gmail.com (Z.S.); ronmarom@gmail.com (R.M.); snirdekalo@gmail.com (S.D.); roymano78@gmail.com (R.M.); ofer@oy-urology.co.il (O.Y.); 2Pathology Department, Tel Aviv Sourasky Medical Center, Tel Aviv 6423906, Israel; ibrahimf@tlvmc.gov.il; 3Pathology Department, Rabin Medical Center, Petah Tikva 4941492, Israel; rababnaamneh@gmail.com

**Keywords:** incontinence, radical prostatectomy, neuromuscular features

## Abstract

Urinary incontinence after robotic-assisted radical prostatectomy (RARP) has been associated with older age, a longer operative time, a higher BMI, a short membranous urethral length and preoperative erectile function. The authors sought to assess the association between the neuromuscular characteristics and postoperative urinary incontinence. Methods: RARP specimens from 29 men who underwent bilateral nerve sparing were reanalyzed. Urinary incontinence was evaluated using the International Consultation on Incontinence Questionnaire—Short Form (ICIQ-SF) at 6 weeks post surgery and last follow-up. Linear and logistic regression analyses were performed to assess neuromuscular characteristics and incontinence. Results: At the 1-year follow-up, 11 patients (38%) reported severe incontinence (>12 ICIQ-SF score). The median number of peripheral nerves observed at the base and apex in the specimens was 52 (IQR 13–139) and 59 (IQR: 28–129), respectively. Ganglia were present in 19 patients (65%) at the base and 12 patients (41%) at the apex. Additionally, the median proportional area of detrusor smooth muscle fibers at the base was 0.54 (IQR 0.31–1), while the median proportional area of striated muscle fibers at the apex was 0.13 (IQR 0.08–0.24). No statistically significant associations were found. Conclusions: Histologic neuromuscular characteristics were not associated with postoperative urinary incontinence. Enhanced intraoperative evaluation and larger-scale studies may prove useful for the prediction of postprostatectomy incontinence.

## 1. Introduction

Prostate cancer is the second most common cancer among men globally, affecting approximately 1.1 million men annually [1]. Robotic-assisted radical prostatectomy (RARP) is considered a major treatment alternative for intermediate and high-risk prostate cancer. Accumulating experience over the past two decades has led to improved postoperative outcomes in terms of blood loss, transfusion rates, nerve sparing, the recovery of urinary continence and erectile dysfunction (ED) [2,3]. Male urinary incontinence (UI) after radical prostatectomy (RP) is predominantly iatrogenic. It is primarily stress UI, characterized by involuntary leakage during physical effort, exertion, sneezing or coughing [4]. Most patients experience transient incontinence after RP, with significant improvements typically achieved within 2–3 months [5]. Despite substantial progress in surgical techniques, the prevalence of postoperative incontinence remains high, estimated to range from 2% to 66%. Studies report continence rates of 68% to 97% at 12 months, with further improvements up to 2 years post-surgery [6,7,8,9,10]. Numerous studies have established potential predictors of postoperative UI, namely increased patient age, a longer operative time, extensive dissection during surgery, a higher body mass index (BMI), a shorter membranous urethral length, injuries to the supporting structures of the urethra, lesions or damage to the neurovascular bundle (NVB) or even detrusor underactivity, the development of postoperative fibrosis and preoperative erectile dysfunction [11,12,13,14,15].

Postoperative urinary incontinence is one of the significant complications of RARP and affects the patient’s quality of life. Although many studies have investigated multiple factors as the causes of urinary incontinence, nerve sparing is one of these factors. There are three types of nerve-sparing techniques, intra-, inter- and extra-prostatic, with differences in oncological and functional outcomes. The more extended the dissection is, the more the oncological outcomes are improved, and the continence is poorer [16,17,18,19,20]. Patients treated at high-volume centers by experienced surgeons are more likely to achieve continence [21]. While some have suggested that the adequate preservation of the neurovascular bundles may expedite the recovery of postoperative urinary continence [22,23,24,25], the relationship between the two remains elusive [26].

There are no reports that have quantified the amount of residual nerve tissue (as inferred from the nerve tissue attached to the removed prostatic tissue) and examined its relationship with urinary incontinence. In this study, the authors examined the relationship between the state of nerve tissue attachment and postoperative urinary incontinence, focusing on the removed prostate tissue. The objective was to quantify the amount of retained neurovascular tissue on radical prostatectomy specimens as a surrogate for the quality of neurovascular bundle preservation and assess its association with the recovery of postoperative UI.

## 2. Materials and Methods

After institutional review board approval (0525-22-TLV), our departmental database was queried to retrieve the medical records of 50 consecutive male patients who underwent bilateral nerve sparing RARP between October 2021 and March 2022. All procedures were performed by three expert surgeons who had already reached a plateau in their learning curves [27,28]. The decision to perform extended pelvic lymph node dissection and/or a nerve-sparing technique during RARP depended on the baseline characteristics of the patients and the tumor characteristics, in accordance with the American Urological Association (AUA)/American Society for Radiation Oncology (ASTRO)/Society of Urologic Oncology (SUO) recommendations [29]. At the end of each procedure, the authors routinely placed a Jackson–Pratt pelvic drain forming a ‘U’ shape anterior to the bladder and the anastomosis. Postoperative management included clinical and laboratory follow-ups, input and output measurements, enhanced recovery protocols and routine discharge on postoperative day (POD) 2 with a urinary catheter in place after the removal of the drain. The urinary catheter was routinely removed at the outpatient clinic on POD10. The exclusion criteria included patients who underwent transurethral prostatectomy (TURP) and those with prior prostate or bladder radiation, known neurological diseases or uncontrolled diabetes. Based on their clinical evaluations and disease features, a bilateral intrafascial dissection technique was employed in all patients [30]. Clinical and pathological data were collected and reviewed, including age and comorbidities, performance status, prostate-specific antigen (PSA) levels and imaging (MRI and PET-PSMA) findings. Pathological analysis included the tumor stage and grade (TMN), margin status and tumor size.

Urinary functional outcomes were assessed at 2 consecutive points in time during follow-up: immediately (6 weeks) after surgery and at the last follow-up. The former included the number of pads used per day and the IPSS score, whereas the latter involved a formal interview where patients were asked to complete the International Consultation on Incontinence Questionnaire—Short Form (ICIQ-SF) [31]. The ICIQ-SF is composed of 3 questions: question 1 (Q1) assesses the frequency of urinary leakage, question 2 (Q2) evaluates the amount of leakage and question 3 (Q3) measures the extent to which UI affects daily life. The scoring categories for incontinence are further stratified into slight (1–5), moderate (6–12), severe (13–18) and very severe (19–21). Of the 50 patients, 29 responded and provided consent to be included in the study.

A dedicated genitourinary pathologist reanalyzed all RP specimens, evaluating the neuromuscular characteristics at the base and apex of the specimen. Hematoxylin and eosin slides of the radical prostatectomy specimens were scanned using the Philips UFS scanner (Koninklikje Philips, Amsterdam, The Netherlands). Skeletal muscle and smooth muscle tissue was labeled on the digital slides by two pathologists using the closed freeform annotation tool in the Philips digital pathology system, which automatically calculates the area of the annotation. Six explicit neuromuscular features were investigated, including (1) the presence of ganglia at the base of the prostate, (2) the presence of ganglia at the apex of the prostate, (3) the proportional area of detrusor muscle fibers at the base, (4) the proportional area of sphincteric striated muscle fibers at the apex, (5) the number of nerves at the base and the (6) number of nerves at the apex (Figure 1). A score was generated for each specimen based on these findings, and its relationship with the severity of incontinence was analyzed.

Descriptive statistics were used to summarize the patients’ characteristics. Continuous variables were reported as the median and inter-quartile range (IQR), and categorical variables were reported as proportions (%). Linear and logistic regression analyses were conducted to assess the relationship between the specimen neuromuscular characteristics and the severity of UI, controlling for clinical and pathological variables including age, prostate size and pathological stage. All statistical analyses were two-sided, and significance was defined as *p* < 0.05. The SPSS software (IBM SPSS Statistics, Version 25, IBM Corp., Armonk, NY, USA) was used.

## 3. Results

### 3.1. Post-Surgery Continence

Of the 50 patients who underwent RARP in our medical facility between October 2021 and March 2022, 29 responded and provided consent to be included in the study. At 6 weeks post-surgery, seven patients (24%) reported full urinary continence (day and night continence and no protective pads used). At the 1-year post-surgery follow-up, patients were interviewed and stratified into three groups based on their ICIQ scores. Nine patients (31%) were categorized as slight UI (ICIQ-SF score < 6), nine patients (31%) were categorized as moderate UI (ICIQ-SF scores 6–12) and 11 patients (38%) were categorized as severe and very severe UI (ICIQ-SF score > 12). There was no statistically significant association between the early continence rate and the severity of UI at the last follow-up (*p* = 0.182). Of the 12 patients with adequate preoperative potency, the recovery of erectile function 1 year after surgery was documented in three based on their IIEF scores (Table 1).

### 3.2. ICIQ Score and Patients’ Disease Features

Based on the median and average ICIQ scores, the patients were restratified into two groups: (1) ICIQ scores 1–12—slight to moderate UI—average score of 6.5 and (2) ICIQ scores 13–21—severe to very severe UI—average score of 16.4. While there was no difference in the patients’ average age between the groups, at 67.3 vs. 69.4, respectively, a statistically significant correlation was found between the severity of incontinence and the risk of disease progression. Compared to patients with lower ICIQ scores, those with higher ICIQ scores had significantly higher preoperative PSA levels, at 9.1 vs. 16. However, the number of patients with preoperative lower urinary tract symptoms defined as IPSS > 7, the prostate size, the pathological stage and the early continence rates were not different between the groups (Table 2).

### 3.3. Neuromuscular Characteristics

There was no significant correlation observed between the degree of early (6 weeks postoperative) or late (1 year) urinary incontinence and any of the tested neuromuscular histologic features of the specimens (Table 3 and Figure 1).

## 4. Discussion

In this study, the authors evaluated the urinary functional outcomes and potential pathological predictors of incontinence in 29 patients who underwent bilateral nerve-sparing RARP.

The early continence rates did not predict the long-term severity of incontinence. Additionally, the authors did not find any significant correlation between the degree of early (6 weeks postoperative) or late (1 year) urinary incontinence and any of the examined neuromuscular histologic features of the specimens.

Our findings demonstrate that postoperative incontinence was more common among higher-risk tumors; however, no associations were found between the pathological neuromuscular characteristics of the specimens and postoperative UI.

The rapid return of urinary control is a critical step in achieving overall satisfaction after radical prostatectomy [2,22]. UI is considered one of the most distressing side effects following surgery and strategies to improve the continence outcomes have been a focus of research in this field [9,10,23]. While the importance of neurovascular preservation to improve early postoperative urinary continence has been demonstrated repeatedly [32], most studies have evaluated the adequacy of nerve sparing based on subjective surgeon-reported notes. Within this context, the authors endeavored to study the relationship between the amount of neurovascular tissue found on the pathological specimen in men undergoing bilateral nerve preservation as a surrogate for inadequate nerve sparing and possibly postoperative urinary incontinence. The addition of useful information to a standard pathology report might potentially allow a more aggressive approach to treating postprostatectomy incontinence.

Traditionally, intraoperative damage to the external urethral sphincter or its innervation was considered the main cause of incontinence after radical prostatectomy [33]. However, the true mechanism has not been completely understood, and, currently, it is thought to be multifactorial. Factors potentially affecting post-surgery incontinence include the patient’s age and comorbidities, obesity, preoperative lower urinary tract dysfunction, the prostate size, the membranous urethra length, bladder neck preservation and the preservation of the membranous urethra-supporting structures [33]. Furthermore, bladder dysfunction can be affected by bladder mobilization during RALP. This is potentially due to several factors: partial somatic and autonomic decentralization, inflammation or infection and geometric alterations of the bladder wall associated with pre-existing hypoxemia [34,35,36].

A urodynamic study before and after RP reported that the maximum urethral closure pressure at rest immediately after RP was reduced to roughly 40% of its preoperative level [37,38,39]. At 1 year postoperatively, the maximal urethral closure pressure had improved, but not to its preoperative point [37,38,39].

The presence of neuromuscular tissue on the outer surface of the prostate specimen might not just be an anatomical observation. It holds potential significance in predicting functional outcomes, including the recovery of urinary continence [40]. While a preserved denser nerve network might theoretically offer better functional outcomes, its presence on the excised specimen suggests that it was not adequately secured during surgery, which might be deleterious [41]. Moreover, the observed association between the cancer stage and incontinence might reflect the tendency to sacrifice, at least in part, the neurovascular tissue surrounding the gland in locally advanced tumors [42]. Kaye et al. have shown that sparing at least one neurovascular bundle along with its supportive tissue has a dramatic effect on the recovery of urinary continence and quality of life in preoperatively potent men [43]. The cavernous nerve (CN) is the main autonomic nerve regulating penile erection but is also involved in the voiding reflex by innervating the urethral transverse muscle on the anterior aspect of the prostate through designated branches [44].

Meticulous nerve preservation during radical prostatectomy has been linked with enhanced postoperative erectile function outcomes [45]. However, the relationship between nerve sparing and urinary incontinence is less clear, pointing towards possible overlapping anatomical or physiological pathways [46]. Due to its small size and the small number of men with documented erectile function recovery, our study could not corroborate the previously observed association between the two. Nonetheless, in the absence of an association between the retained specimen neuromuscular components and urinary function outcomes, our findings might suggest that a non-nerve-sparing operation in locally advanced tumors (extrafascial dissection) might not be inevitably deleterious for the rapid recovery of urinary continence.

Several limitations need to be emphasized. First, the high prevalence of preoperative erectile dysfunction in our study impeded our ability to study the association between nerve sparing, postoperative ED, and UI. Second, due to the retrospective nature of this study, the authors were unable to calculate the weight of used pads, obtain bladder diaries, or perform urodynamic assessments, which are valuable tools for more accurate assessments. Additionally, the modest cohort size might have limited our ability to provide definitive proof of an association between the neuromuscular specimen characteristics and urinary incontinence [47]. There is clear need for more extensive research to better comprehend these interconnections and improve surgical practices and patient outcomes [48].

## 5. Conclusions

Histologic neuromuscular characteristics in radical prostatectomy specimens were not associated with postoperative continence outcomes. Further studies that incorporate additional assessments, such as myelin staining, might provide different insights regarding the integrity of the neurovascular bundles and of its association with incontinence outcomes.

## Figures and Tables

**Figure 1 diagnostics-14-02001-f001:**
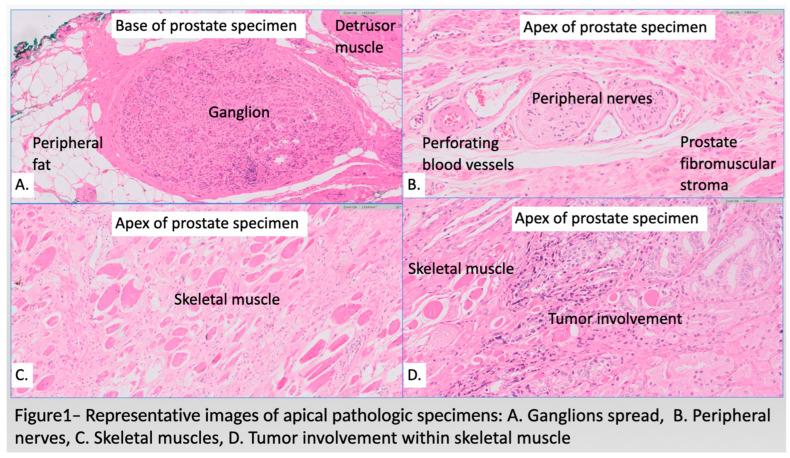
Representative images of pathologic specimens: (**A**) Ganglia spread, X10 multiplication (**B**) peripheral nerves, X20 multiplication (**C**) skeletal muscle, X10 multiplication (**D**) tumor involvement within skeletal muscle, X20 multiplication.

**Table 1 diagnostics-14-02001-t001:** Patients’ postoperative continence and erectile preservation based on ICIQ score.

ICIQ Category (12 Months Post-Surgery)	Number of Patients (Percentage)	Early Full Continence—Number of Patients (6 Weeks Post-Surgery) (Percentage from the Group)	Erectile Preservation (Number of Patients with Preoperative Erectile Function)
Slight (1–5)	9 (31%)	4 (44%)	2 (4)
Moderate (6–12)	9 (31%)	2 (22%)	1 (4)
Severe and very severe (13–21)	11 (38%)	1 (9%)	0 (4)

ICIQ = International Consultation on Incontinence Questionnaire.

**Table 2 diagnostics-14-02001-t002:** Patients’ disease features stratified according to ICIQ score at 1 year following RARP.

	ICIQ ≤ 12 (n = 18)	ICIQ > 12 (n = 11)	*p* Value
Average ICIQ (IQR)	6.5 (2.25–10.75)	16.4 (14.5–18)	
Age (IQR)	67.3 (63.9–73.3)	69.4 (67–72.7)	0.44
Preoperative PSA (ng/mL) (IQR)	9.1 (6.6–9.9)	16 (9–22.2)	0.02
Number of patients with preoperative LUTS (IPSS > 7)	8 (45%)	5 (45%)	0.96
ISUP score			0.302
1	1 (5.5%)	0 (0%)	
2	8 (44.4%)	3 (27.2%)	
3	5 (27.7%)	4 (36.3%)	
4	2 (11.1%)	2 (18.1%)	
5	2 (11.1%)	0 (0%)	
		ISUPx—after hormonal treatment—2 (18.1%)	
Prostate size (gram)	68 (55–73)	63.2 (49.5–65.3)	0.6
pT stage (2009)			0.136
T2	8 (44.4%)	1 (9%)	
T3a	9 (50%)	9 (82%)	
T3b	1 (5.5%)	1 (9%)	
Full 6W continence day	8 (45%)	3 (27%)	0.37
Full 6W continence night	8 (45%)	2 (18%)	0.14

RARP = robotic-assisted radical prostatectomy, ICIQ = International Consultation on Incontinence Questionnaire, LUTS = lower urinary tract symptoms, IPSS = International Prostate Symptom Score, ISUP score = International Society of Urological Pathology, pT stage = pathological T stage.

**Table 3 diagnostics-14-02001-t003:** Neuromuscular characteristics of pathologic specimens stratified by incontinence severity.

	Overall (n = 29)	ICIQ Score ≤ 12 (n = 18)	ICIQ Score > 12 (n = 11)	*p* Value
Presence of ganglion in the base, n (%)	19 (65%)	11 (61%)	8 (72%)	0.22
Presence of ganglion in the apex, n (%)	12 (41%)	8 (44%)	4 (36%)	0.11
Proportional area of detrusor muscle in the base, median (IQR)	0.54 (0.31–1.00)	0.45 (0.29–0.92)	0.60 (0.40–1.00)	0.57
Proportional area of striated muscle at the apex, median (IQR)	0.13 (0.07–0.24)	0.13 (0.06–0.24)	0.13 (0.08–0.24)	1.00
Number of nerves at the base, median (IQR)	52 (13–139)	69 (11–152)	36 (14–108)	0.65
Number of nerves at the apex, median (IQR)	59 (28–129)	61 (28–142)	51 (33–116)	0.46

## Data Availability

The data that support the findings of this study are available on request from the corresponding author, Tomer Bashi. The data are not publicly available due to ethical issues and the privacy of the participants.

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
