# Peer review of "The Association between Specimen Neuromuscular Characteristics and Urinary Incontinence after Robotic-Assisted Radical Prostatectomy"

_diagnostics, 2024, doi:10.3390/diagnostics14182001_

Round 1

Reviewer 1 Report (Previous Reviewer 4)

Comments and Suggestions for Authors

Please check the following items.

1. Reference is written in the introduction section.

2. The introduction is redundant. Please write more briefly. Postoperative urinary incontinence is one of the significant complications of RARP and affects the patient's quality of life. Although many studies have investigated multiple factors as causes of urinary incontinence, nerve sparing is one of these factors. There are three types of nerve-sparing techniques: intra-, inter-, and extra-prostatic, but there are no reports that have quantified the amount of residual nerve tissue (as inferred from the nerve tissue attached to the removed prostatic tissue) and examined its relationship with urinary incontinence. In this study, the authors examined the relationship between the state of nerve tissue attachment and postoperative urinary incontinence, focusing on the removed prostate tissue. It is sufficient to simply state this description.

3. The explanation of ICIQ is repeated in the main text. Please organize it.

4. The "However~" description in the result section 3.2 is unclear. Please re-write.

5. The description of the evaluation items in Results 3.3 is redundant with the description in the Methods section, so I think it is unnecessary. Using "we" as the subject in the main text does not give a good impression of a scientific paper.

Comments on the Quality of English Language

There are some spellings missing in the text.

Author Response

Response to reviewer: 

Comment - 

1. Reference is written in the introduction section.

2. The introduction is redundant. Please write more briefly. Postoperative urinary incontinence is one of the significant complications of RARP and affects the patient's quality of life. Although many studies have investigated multiple factors as causes of urinary incontinence, nerve sparing is one of these factors. There are three types of nerve-sparing techniques: intra-, inter-, and extra-prostatic, but there are no reports that have quantified the amount of residual nerve tissue (as inferred from the nerve tissue attached to the removed prostatic tissue) and examined its relationship with urinary incontinence. In this study, the authors examined the relationship between the state of nerve tissue attachment and postoperative urinary incontinence, focusing on the removed prostate tissue. It is sufficient to simply state this description.

Response - 

Thank you for the comment. According to the reviewer's comment we shortened the introduction section. However, we had to balance it against other reviewer's comment to supply sufficient background and overview of the current knowledge. All references of the introduction are provided in [..]. 

Changes made - introduction: 

"Prostate cancer is the second most common cancer among men globally, affecting approximately 1.1 million men annually [1]. Robotic-assisted radical prostatectomy (RARP) is considered a major treatment alternative for intermediate and high-risk prostate cancer.  Accumulating experience over the past 2 decades has led to improved postoperative outcomes in terms of blood loss, transfusion rate, nerve sparing, recovery of urinary continence and erectile dysfunction (ED) [2, 3]. Male urinary incontinence (UI) after radical prostatectomy (RP) is predominantly iatrogenic. It is primarily stress UI, characterized by involuntary leakage during physical effort, exertion, sneezing, or coughing [4]. Most patients experience transient incontinence after RP, with significant improvement typically achieved within 2–3 months [5]. Despite substantial progress in surgical technique, the prevalence of postoperative incontinence remains high, estimated to range from 2% to 66%. Studies report continence rates of 68% to 97% at 12 months, with further improvements up to 2 years post-surgery [6-10]. Numerous studies have established potential predictors of postoperative UI namely increased patient age, longer operative time, extensive dissection during surgery, higher body mass index (BMI), shorter membranous urethral length, injuries to the supporting structures of the urethra, lesions or damage to the neurovascular bundle (NVB) or even detrusor underactivity, development of postoperative fibrosis and preoperative erectile dysfunction [11-15].  

Postoperative urinary incontinence is one of the significant complications of RARP and affects the patient's quality of life. Although many studies have investigated multiple factors as causes of urinary incontinence, nerve sparing is one of these factors. There are three types of nerve-sparing techniques: intra-, inter-, and extra-prostatic, with differences in oncological and functional outcomes. The more extended the dissection is, the oncological outcomes are improved, and continence is poorer [16-20]. Patients treated at high-volume centers by experienced surgeons are more likely to achieve continence [21]. While some suggested that adequate preservation of the neurovascular bundles may expedite the recovery of postoperative urinary continence [22, 23, 24, 25], the relationship between the two remains elusive [26].

There are no reports that have quantified the amount of residual nerve tissue (as inferred from the nerve tissue attached to the removed prostatic tissue) and examined its relationship with urinary incontinence. In this study, the authors examined the relationship between the state of nerve tissue attachment and postoperative urinary incontinence, focusing on the removed prostate tissue. The objective was to quantify the amount of retained neurovascular tissue on radical prostatectomy specimens as a surrogate for the quality of neurovascular bundles preservation and assess its association with recovery of postoperative UI."

Comment - 

3. The explanation of ICIQ is repeated in the main text. Please organize it.

Response - 

According to the comment, we organized the explanation of ICIQ to be more clear.

Changes made - methods:

"The ICIQ-SF is comprised of 3 questions: Question 1 (Q1) assesses the frequency of urinary leakage, question 2 (Q2) evaluates the amount of leakage, and question 3 (Q3) measures the extent to which UI affects daily life. The scoring categories for incontinence are further stratified into slight (1-5), moderate (6-12), severe (13-18), and very severe (19-21). Of the 50 patients, 29 responded and provided consent to be included in the study."

Comment - 

4. The "However~" description in the result section 3.2 is unclear. Please re-write.

Response - 

We agree with the reviewer that this is unclear. We edited it to improve it. Thank you for the comment. 

Changes made - results section:

"However, number of patients with preoperative lower urinary tract symptoms defined as IPSS>7, prostate size, pathological stage and early continence rates were not different between the groups (table 2)."

Comment - 

5. The description of the evaluation items in Results 3.3 is redundant with the description in the Methods section, so I think it is unnecessary. Using "we" as the subject in the main text does not give a good impression of a scientific paper.

Response - 

According to the comment, we omitted the description of the evaluation items. In addition, we replaced the word "we" with "the authors". 

Changes made - results section:

"The authors did not find any significant correlation between the degree of early (6-weeks postoperative) or late (1-year) urinary incontinence and any of the tested specimen neuromuscular histologic features.  (Table 3 and Figure 1)."

Attached is the revised manuscripts with the corrections marked in yellow.

Reviewer 2 Report (Previous Reviewer 3)

Comments and Suggestions for Authors

Accepted in the re-submitted form

Author Response

Thank you.

Reviewer 3 Report (Previous Reviewer 2)

Comments and Suggestions for Authors

The required revisions were performed. The article was significantly improved. It may be considered for publication.

Author Response

thank you

Round 2

Reviewer 1 Report (Previous Reviewer 4)

Comments and Suggestions for Authors

Writing suggestions about the sentence not including 'We' or 'the authors.'

The purpose of this study was to investigate the relationship between the state of nerve tissue attachment and postoperative urinary incontinence, focusing on the removed prostate tissue.

No significant correlation was found between the degree of early (6 weeks post-surgery) or late (1 year) urinary incontinence and the neuromuscular histological characteristics of the specimens examined. (Table 3 and Figure 1).

I think you can delete the following sentences from the Discussion section. ‘’In this study, the authors evaluated the urinary functional outcomes and potential pathological predictors of incontinence in 29 patients who underwent bilateral nerve-sparing RARP. ’’

Author Response

comment 1 - Writing suggestions about the sentence not including 'We' or 'the authors.'

response- 

There was no significant correlation observed between the degree of early (6-weeks postoperative) or late (1-year) urinary incontinence and any of the tested specimen neuromuscular histologic features.  (Table 3 and Figure 1).

comment 2 - I think you can delete the following sentences from the Discussion section. ‘’In this study, the authors evaluated the urinary functional outcomes and potential pathological predictors of incontinence in 29 patients who underwent bilateral nerve-sparing RARP. ’’

response - Thank you for the comment.  We added this section due to several reviewers comments requesting this background section.

This manuscript is a resubmission of an earlier submission. The following is a list of the peer review reports and author responses from that submission.

Round 1

Reviewer 1 Report

Comments and Suggestions for Authors

Thank you for inviting me to review this manuscript.

In this retrospective study, the authors analyze the association between neuromuscular characteristics of specimens and urinary incontinence after Robotic-Assisted Radical Prostatectomy.

I found the study design interesting; nevertheless, I encountered some flaws that I aim to discuss with the authors.

I apologize for my "copy and paste" approach, but the manuscript lines were not enumerated, making it difficult to refer to specific parts.

ABSTRACT

Radical prostatectomy is usually abbreviated as RP. RARLP is not commonly used; RARP is preferred for Robotic-Assisted Radical Prostatectomy. Please choose one abbreviation and use it consistently throughout the manuscript.

INTRODUCTION

This section does not provide a sufficient overview of the background and current knowledge. It should better contextualize the current work and introduce it more effectively.

METHODS

"50 consecutive male patients who underwent bilateral nerve-sparing RALRP between October 2021 and March 2022": Why did you only consider a six-month period?

The authors did not mention the surgical technique, the surgeons, their expertise, or established exclusion criteria (e.g., previous pelvic surgery, previous TURP, previous radiotherapy, adjuvant radiotherapy, etc.).

RESULTS

All results presented in the tables should also be presented in the main text.

"At a median follow-up of 11 months": Follow-up cannot be described only as a median. Additionally, if the median was 11 months, this would mean that some patients had a follow-up of less than 11 months, which is insufficient to evaluate continence recovery after radical prostatectomy.

DISCUSSION

This section needs significant improvement. I suggest a more structured format. All discussed points should first appear in the results (e.g., Our findings demonstrate that post-operative incontinence was more common among higher-risk tumors).

CONCLUSIONS

These should be rewritten once the results and discussion are appropriately revised.

Overall, I suggest revising the text for better clarity. Acronyms should be used consistently for terms like radical prostatectomy, nerve sparing, urinary incontinence, etc.

Comments on the Quality of English Language

The use of English in the manuscript is generally good, requiring only minor adjustments to improve conciseness and a few isolated grammatical corrections.

Reviewer 2 Report

Comments and Suggestions for Authors

It is a study investigating the correlation between the NVB specimens in the excised prostate and the postoperative continence and potency status. Despite the innovative idea of the authors, there are many concerns. 

Comment 1: The introduction section should focus on the factors affecting postoperative continence and potency status. The different techniques of NVB preservations should also mentioned. 

Comment 2: The Study desing is not clearly described. A comparison between pre- and postoperative IIEF-5 score may be beneficial. There is no information concerning the presence of postoperative complications or not. There is no information regarding the hospitalization and catheterization duration. There is no defined subgrouping.

Comment 3: Despite the fact that the main purpose of the study is the correlation between the specimens of NVB and the postoperative potency and continence status, the results section focus on the ICIQ-SF subgroups.

Comment 4: The discussion section should be enriched.

Comments on the Quality of English Language

As described above

Reviewer 3 Report

Comments and Suggestions for Authors

Dear authors,

Thank you for your creative manuscript. However, there are some issues to be discussed before a possible publication.

- you are exploring the functional outcomes after RALRP, but you are using only subjective scores. Even used pads is considered as subjective, as you do not request documentation of their weight difference before and after use. You have to use bladder diaries, in order to provide sufficient results.

- you do not provide any data about functional disorders before surgery, so any relevant outcome after RALRP cannot be considered as absolutely reliable, as it cannot be compared to any baseline.

- trying to explore neuromascular disorders in the clinical practice, you need urodynamics, as this is the only way to describe the bladder behavor before and after surgery. Please, provide any available data.

Reviewer 4 Report

Comments and Suggestions for Authors

This retrospective observational study examined 29 cases of robot-assisted radical prostatectomy (with nerve-sparing techniques). Postoperative urinary continence condition was evaluated at two time points, the first being 6 weeks after surgery (early period), and the second was not standardized, with a median of 11 months (late period). At the second evaluation time point, there was no statistical association between the condition of urinary continence and the histopathological characteristics of the prostate (in particular, the details of the neuromuscular tissue).

To better understand this study, this reviewer would like to make some points for improvement.

Robotic-assisted prostatectomy is generally described as RARP. I don't think RALRP is common.

Please include the institutional approval number for the study for accuracy.

I recommend changing the Results section to be organized and described in subheadings. For example, â‘ patient background, â‘¡continence at the first evaluation, â‘¢ICIQ results at the second evaluation, â‘£factors examined from the two groups based on ICIQ results

Patients were divided into two groups based on the mean or median ICIQ score at the second evaluation time point, 11 months after surgery, but I don't understand what this means. As mentioned in the Methods section, the ICIQ classification is divided into moderate or less and severe or more based on a score of 12 points, right? From this, I think it means that the 29 patients were divided into two groups based on an absolute value of 12 points (not mean or median score), right?

The information in Table 2 is insufficient. When using abbreviations in a table, please include an explanation. Shouldn't the two cases that received preoperative hormone therapy be excluded from this study, considering the possibility that it could affect the pathological tissue results? What does the notation "pT stage 2009" mean? What does the notation "ISUP score" mean? Please describe it in a way that any reader can understand.

There is a lack of information regarding the most important part of this study, the histopathological evaluation. You should include details of the method used to calculate the muscle area.

The study hypothesized that if more nerve tissue was preserved, postoperative continence would be achieved. In other words, cases where there was a large number of ganglia attached to the prostate tissue and a large number of nerve fibers remaining were judged to have "insufficient" nerve preservation. In this study, the cases with more severe incontinence had fewer ganglia and fewer nerve fibers. In other words, it was judged that nerve preservation had been "sufficiently performed". The hypothesis and the results were reversed. What do you think about that?

Comments on the Quality of English Language

none